# SCALING FAIR LEARNING TO HUNDREDS OF INTERSECTIONAL GROUPS

## ABSTRACT

Bias mitigation algorithms aim to reduce the performance disparity between different protected groups. Existing techniques focus on settings where there is a small number of protected groups arising from a single protected attribute, such as skin color, gender or age. In real-world applications, however, there are multiple protected attributes yielding a large number of intersectional protected groups. These intersectional groups are particularly prone to severe underrepresentation in datasets. We conduct the first thorough empirical analysis of how existing bias mitigation methods scale to this setting, using large-scale datasets including the ImageNet People Subtree and CelebA. We find that as more protected attributes are introduced to a task, it becomes more important to leverage the protected attribute labels during training to promote fairness. We also find that the use of knowledge distillation, in conjunction with group-specific models, can help scale existing fair learning methods to hundreds of protected intersectional groups and reduce bias. We show on ImageNet's People Subtree that combining these insights can further reduce the bias amplification of fair learning algorithms by 15% —a surprising reduction given that the dataset has 196 protected groups but fewer than 10% of the training dataset has protected attribute labels.

## 1 INTRODUCTION

In many real-world applications, there is a significant potential for harm when the predictive properties and performance of machine learning models vary across different demographic populations. This is indeed the case for many applications including facial analysis (Buolamwini & Gebru, 2018), ad delivery (Sweeney, 2013) and search engines (Noble, 2018). Recent works aim to address this by designing learning algorithms that guarantee or approximate formal notions of algorithmic fairness (Mehrabi et al., 2021). However, there still remains a gap between theory and practice.

This gap is especially pronounced in the case of *intersectional fairness*. In the real-world, it is critical to simultaneously protect multiple attributes such as gender, skin color and age. Due to increasing disadvantage along the intersecting dimensions of protected attributes (Crenshaw, 1989; Foulds et al., 2020), it is also important to protect the intersections of these attributes. However, there has been scarce work on intersectional and subgroup fairness in deep learning contexts, with a large portion of the literature still focusing on single (binary) protected attribute settings, such as protecting the Male/Not-Male attribute in the CelebA Dataset (Liu et al., 2015).

In this work, we focus on intersectional fairness and seek to scale deep learning bias mitigation techniques to multiple protected attributes with hundreds of intersectional groups. Approaches such as training separate classifiers for each group (Wang et al., 2020), upweighting the losses of samples based on the group size, optimizing for the worst-case group outcome (Sagawa et al., 2020), or upweighting errors in early epochs (Liu et al., 2021) have all been shown to be effective in the single attribute setting. However, it is not clear whether they are effective given the intersection of multiple protected attributes, where there is a combinatorial number of protected groups and fairness for individual protected attributes does not imply intersectional fairness (Figure 1b). We address the question of how to scale existing bias mitigation techniques to this important but challenging setting.

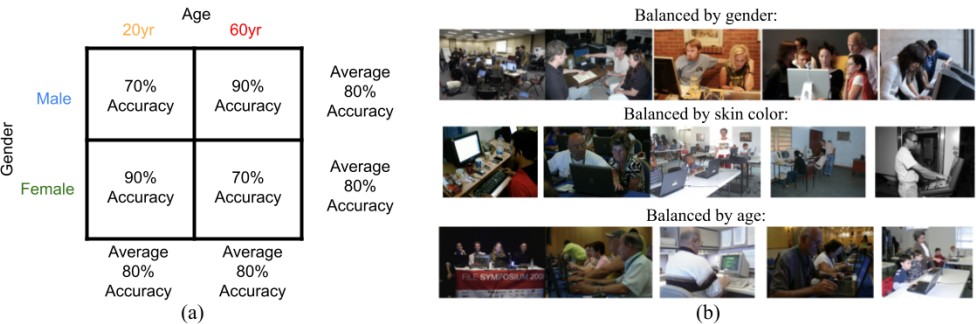

Figure 1: (a) A toy example of intersectional bias. Suppose all intersectional groups are equally represented in the data. A model has the same 80% average accuracy on each of the listed age groups and genders. However, some intersectional groups (e.g., 60 yr females) have only 70% accuracy. (b) Figure of samples from the ImageNet's "Programmer" synset but balanced along the attributes of gender, skin color and age (taken from Yang et al. (2020)). Our ImageNet experiments explore methods of mitigating bias against the intersection of these attributes.

**Summary of Contributions:**

1. We perform a thorough empirical analysis of existing deep learning bias mitigation techniques to explore their applicability in multi-attribute fairness settings. We find that as the number of protected attributes ($k$) increases, attribute labels become necessary to mitigate biases.

2. We conduct the first study of bias mitigation on the ImageNet People Subtree, with 196 protected groups but fewer than $10\%$ of training data having protected attribute labels. In this challenging setting, we show existing bias mitigation methods can reduce bias amplification (Zhao et al., 2017) by 9% against empirical risk minimization (ERM). We identify two key issues with scaling up to so many intersectional groups: model complexity and overfitting to limited attribute labels.

3. We address these scaling challenges by proposing a novel regularization method: Knowledge **D**istillation of **I**ndependent Models as **R**egularization (DIR). DIR regularizes a single student classifier with teacher classifiers trained specifically for each intersectional group to implicitly incorporate attribute label information whenever it's available. In contrast to existing techniques with complexity linear in the number of protected groups, DIR has a constant inference complexity and model size when used as a stand-alone bias mitigation algorithm. We further demonstrate that DIR can regularize other bias mitigation algorithms, reducing bias amplification by an additional 15% on ImageNet.

We first conduct a thorough evaluation of existing bias mitigation algorithms through a series of fair learning tasks on the CelebA dataset (Liu et al., 2015). Each round introduces a new protected attribute and evaluates whether existing methods can protect the resulting intersectional groups. Our findings demonstrate that, even with hundreds of intersectional groups, there are bias mitigation algorithms which can significantly reduce bias. However, they all require access to protected attribute labels during training. While bias mitigation methods which do not require attribute labels are occasionally effective in single-attribute settings, as suggested in prior literature (Liu et al., 2021; Shrestha et al., 2021), we show they are not effective in multi-attribute scenarios.

We then consider the more challenging task of learning a fair classifier on the ImageNet People Subtree, which Yang et al. (2020) labels for 196 intersectional groups. This setting is especially challenging for two reasons. First, our previous results suggest that access to protected attribute labels during training is necessary to effectively protect intersectional groups. However, such labels are scarce on our ImageNet dataset, with less than 10% of training datapoints having such information. This leaves bias mitigation algorithms prone to overfitting on the few available attribute labels. Second, bias mitigation algorithms often have a model size and complexity that scale linearly with the number of protected groups, which, in turn, explode combinatorially due to intersectionality.

We address these challenges by proposing Knowledge Distillation of Independent Models as Regularization (DIR). DIR uses group-specific teacher classifiers to train a single student classifier that, at each datapoint, mimics the output of the teacher classifier corresponding to the datapoint's ground-truth group. DIR, whose model complexity is independent of the number of protected attributes ($k$), is an efficient alternative to existing bias mitigation algorithms which scale linearly with $k$.

DIR also provides a regularization scheme for other bias mitigation techniques, reducing overfitting by implicitly incorporating attribute label information into learning objectives. Using DIR, we can significantly mitigate bias towards intersectional groups on ImageNet, empirically demonstrating a 22% reduction in bias amplification over standard empirical risk minimization.

## 2 RELATED WORK

**Algorithmic Fairness** Prior works have explored algorithmic bias in a number of real-world applications. Deep learning models, including commercial image classifiers (Buolamwini & Gebru, 2018) and natural langauge processing (Alvi et al., 2018; Bolukbasi et al., 2016; Garg et al., 2018), have been the subject of significant scrutiny. Recent literature have proposed methods for bias mitigation (Edwards & Storkey, 2015; Ramaswamy et al., 2021; Ryu et al., 2018; Wang et al., 2020; Zhao et al., 2017), approximately optimizing metrics like worst-case accuracy (weighted on the worst-off group), mean accuracy (balanced over protected groups), bias amplification scores (Zhao et al., 2017), and intersectional bias scores (Foulds et al., 2020). Most build on existing techniques in classical fairness, such as reductions approaches, adversarial models, fairness through awareness, and importance weighting (Agarwal et al., 2018; Saerens et al., 2002; Dwork et al., 2012; Zhang et al., 2018). Many also borrow on techniques from robust learning (Adragna et al., 2020; Liu et al., 2021; Sagawa et al., 2020), causal inference (Arjovsky et al., 2020; Creager et al., 2021; Kusner et al., 2017; Madras et al., 2019), and representation learning (Pezeshki et al., 2020).

**Intersectional Multi-Attribute Fairness** One of our contributions is replicating existing results (Wang et al., 2020; Shrestha et al., 2021; Liu et al., 2021; Sagawa et al., 2020) and extending their empirical analysis to analogous multi-attribute settings. Shrestha et al. (2021) is the most related work of multi-attribute fairness, but only compare existing algorithms against new sets of datasets. In contrast, we focus on the ImageNet and CelebA datasets and observe algorithms as we tune the number of protected attributes (on a fixed dataset). We reach conflicting findings with Shrestha et al. (2021). While methods that do not access protected attribute labels may appear to be effective in certain single-attribute settings, their performance deteriorates in multi-attribute settings. We also find that a reweighting baseline is competitive with methods proposed by Wang et al. (2020). Beyond the deep learning settings considered by our paper and the aforementioned literature, Kang et al. (2021) considers a variational method for multi-attribute fairness in classical learning settings.

**Fairness with Partial Unknown Attributes** In many settings, it may not be practical to label the protected attributes of all datapoints. The setting where only part of the dataset has attribute labels has been explored in previous works: Dai & Wang (2021) targets a specific graph neural network setting and Ho et al. (2020) provides a basic analysis of adversarial fairness. Some works have looked at the case where no labels are available (Chen et al., 2019; Hashimoto et al., 2018). Others have looked at learning proxy labels from a different task (Kallus et al., 2020; Awasthi et al., 2021). Our work focuses on the partial unknown attributes case and provides an in-depth analysis with extensive experiments and larger coverage of methodologies.

## 3 PRELIMINARIES

We formalize our problem as learning a model $h : X \rightarrow Y$, where $X$ is the input space and $Y$ a finite label space. We seek to learn a $h$ that is fair with respect to a $m$-dimensional protected attribute vector $A$ where $m$ is the number of protected attributes. Fairness datasets are thus in tuples $(x, a, y)$ with image $x \sim X$, protected attribute vector $a \in A$, and single/multi-label label $y \in Y$. For example, a protected attribute vector might look like $a = (\text{male}, \text{caucasian}, \text{teenager}, \text{not veteran})$.

For the entirety of this paper, "labeled/unlabeled" refers to whether the protected attribute labels $A$ are available—we always assume the input features $X$ and class labels $Y$ are available. Unlabeled data refers to datapoints where $X$ and $Y$ are available, but $A$ is not.

To be fair with respect to the protected attributes $A$, we want to be fair to the intersectional groups $G$ arising from $A$. The number of such groups is generally combinatorial in $m$. An example is shown in Figure 1(a). We formalize fairness among groups by reviewing two common definitions.

First, demographic parity concerns parity in $\Pr(\hat{y} \mid g)$, the probability of predicting a label $\hat{y} \in Y$ for a datapoint belonging to a group $g \in G$. This definition of fairness requires that, over some fixed

Table 1: List of the bias mitigation algorithms evaluated in Sections 4 & 5.

| Name | Abbreviation | Needs Attribute Labels | Reweighting | Adversarial |
|---|---|---|---|---|
| Empirical Risk Minimization | U-ERM | No | No | No |
| Uniform Reweighting | Weighted ERM | Yes | Yes | No |
| Group D. Robust Optim. (Sagawa et al., 2020) | GDRO | Yes | Yes | Yes |
| Adversarial Censoring (Edwards & Storkey, 2015) | Adversarial | Yes | No | Yes |
| Invariant Risk Minimization (Arjovsky et al., 2020) | IRM | Yes | No | No |
| Domain Independence (Wang et al., 2020) | Independent-SP | Yes | No | No |
| Domain Discriminative (Dwork et al., 2012) | Discriminative | Yes | No | No |
| Spectral Decoupling (Pezeshki et al., 2020) | U-SD | No | No | No |
| Just Train Twice (Liu et al., 2021) | U-JTT | No | Yes | Yes |

data distribution, the predictions of a model $h$ are probabilistically independent of the protected group membership. Formally, $\forall g' \in G, \hat{y} \in Y : \Pr(\hat{y} \mid g) = \Pr(\hat{y} \mid g')$. This objective is generally more applicable to settings where the predicted class $Y$ is a decision (e.g., issuing a loan).

Second, the equalized odds definition refines the demographic parity constraint to enforce parity in $\Pr(\hat{y} \mid g, y)$, the probability of predicting a label $\hat{y} \in Y$, for a datapoint with true label $y \in Y$ belonging to group $g \in G$. This definition enforces, for instance, equivalent accuracies for each group. Formally, $\forall g' \in G, \hat{y}, y \in Y : \Pr(\hat{y} \mid g, y) = \Pr(\hat{y} \mid g', y)$.

## 3.1 Fairness Metrics

Given the difficulty of analyzing neural networks, we use metrics that approximate the margin by which notions of fairness are violated. For equalized odds, two common surrogate metrics include the importance-weighted accuracy and the worst-case accuracy. Importance-weighted accuracy (or "reweighted accuracy") is the accuracy weighted such that each group has equal representation.

$$\text{Acc}_U(h) = \mathbb{E}_{x,y,g \sim (X,Y,G)} \left[ \frac{1}{|g|} \mathbb{1}[y = h(x)] \right] \tag{1}$$

Worst-case accuracy is the accuracy of the group with the lowest accuracy.

$$\text{Acc}_W(h) = \min_{g' \in G} \mathbb{E}_{x,y,g \sim (X,Y,G);g=g'} [\mathbb{1}[y = h(x)]] \tag{2}$$

Similarly, bottom-quartile accuracy is defined as the average of the bottom quartile of group accuracies. Papers based on robust learning often use worst-case accuracy metrics (Sagawa et al., 2020; Liu et al., 2021). However, more general papers also use importance weighted loss, such as Wang et al. (2020). We also define such reweighted and worst-case metrics for Mean Average Precision (mAP), which we use in place of accuracy for multi-label settings.

For demographic parity, two common metrics include bias amplification (Zhao et al., 2017) and intersectional bias (Foulds et al., 2020). Bias amplification measures the difference between a group's ground-truth representation for a label class versus the group's predicted representation:

$$s_y := \Pr(g \mid h(x)) - \Pr(g \mid y) \text{ where } g := \text{argmax}_{g' \in G} \Pr(g' \mid h(x)). \tag{3}$$

A positive score indicates that the model amplifies biases in the data. Generally, we seek a bias amplification score close-to or below zero. However, differences between negative bias amplification scores are non-informative. Bias amplification have been widely adopted by prior works like Wang et al. (2020), but has been found to be limited in what it captures (Wang & Russakovsky, 2021).

The intersectional bias score (Foulds et al., 2020) more directly measures violations of the demographic parity constraint. Larger intersectional bias indicates greater demographic parity violations.

$$\epsilon_y := \max_{g_1, g_2 \in G} \text{argmin}_\epsilon \text{ s.t. } \exp(-\epsilon) \leq \frac{\Pr(h(x) = y \mid g_1)}{\Pr(h(x) = y \mid g_2)} \leq \exp(\epsilon) \tag{4}$$

In contrast to bias amplification, we seek as low an intersectional bias score as possible. However, due to noise, intersectional bias scores can be non-informative when different methods have very similar scores. Since these metrics are defined for each class label $y$, in multi-class and multi-label settings we will occasionally average the metrics over their top quartile of class labels.

Varying Protected Attributes Number on CelebA Blond Hair/Not Blond Task

Figure 2: Bias mitigation algorithms on CelebA-SL task for $k \in [1, 7]$. The left figure plots the reweighted accuracy. Independent-SP and U-JTT have low reweighted accuracy compared to the remaining methods. The middle figure plots bias amplification. Only Independent-SP has a above zero bias amplification score. The right figure plots intersectional bias. U-ERM and U-JTT have poor intersectional bias. The unlabeled method U-SD has competitive intersectional bias for small $k$ but, for large $k$, is outperformed by Weighted ERM with a margin increasing in $k$.

## 3.2 METHODS FOR ANALYSIS

Table 1 lists the bias mitigation algorithms explored in this paper. We consider both "unlabeled" and "labeled" methods, which are distinguished by whether they use protected group information during training. Our definition of "unlabeled" methods still assume access to *class* labels; they instead lack access to *protected attribute* labels. Abbreviations of unlabeled methods are prefixed with a "U-". We include empirical risk minimization ("U-ERM") as a baseline. Other unlabeled methods we consider include spectral decoupling (Pezeshki et al., 2020) and Just Train Twice (Liu et al., 2021).

For labeled methods, we include weighted ERM where the risk in Equation 1 is approximated with a maximum-likelihood-estimate of group importance weights. We also include robust optimization methods that use adversarial importance weights to either maximize the worst-case group accuracy (GDRO) (Sagawa et al., 2020) or eliminate protected group information (Adversarial) (Edwards & Storkey, 2015). We also include invariant risk minimization (IRM) (Arjovsky et al., 2020) for completeness. We include the domain independent and discriminative algorithms (Wang et al., 2020), which train a collection of group-specific classifiers and ensemble them for inference.

## 4 ANALYSIS TOWARDS MULTIPLE PROTECTED ATTRIBUTES

In this section, we analyze how different bias mitigation techniques perform in settings with multiple protected attributes. We build on the CelebA dataset (Liu et al., 2015) experiment settings considered in Wang et al. (2020), Liu et al. (2021), and Shrestha et al. (2021). Specifically, we expand their single-attribute experiments by incrementally introducing additional protected attributes. The Appendix includes additional replication results alongside other competitive baselines.

**Experimental Setup.** We consider two primary experiment settings: (1) a binary classification task, CelebA-SL, on CelebA's blond hair class label, taken from Liu et al. (2021) and Shrestha et al. (2021). (2) a multi-label classification task, CelebA-ML, on 33 out of 40 of CelebA's class labels, taken from Wang et al. (2020). For CelebA-ML, the 7 omitted class labels are reserved as protected attributes and specified in the following paragraph.

For both CelebA-SL and CelebA-ML, we train residual networks (He et al., 2016) using the algorithms listed in Table 1. We evaluate how these methods perform as new protected attributes are introduced by repeating the learning task while each time introducing a new attribute to protect. Let $k$ denote the number of protected attributes. Since every CelebA attribute is binary, every protected attribute that is added doubles the number of intersectional groups. The protected attributes that are added are taken from a fixed set to reduce variance. The pool of potential protected attributes is, in order: Pale Skin, Male, Narrow Eyes, Big Nose, Young, Straight Hair, and Attractive. These at-

Varying Protected Attributes Number on CelebA Multi-Label Task

Figure 3: Bias mitigation algorithms on CelebA-ML task for $k \in [1, 7]$. The left figure plots the reweighted mAP. The left middle plots the bottom quartile mAP scores. U-ERM and Independent-SP score marginally higher than Weighted ERM. The right middle plots bias amplification with Weighted ERM having the lowest bias amplification. The right figure plots intersectional bias with Independent-SP having the lowest intersectional bias. U-JTT starts with low intersectional bias, but grows beyond that of Weighted ERM and Independent-SP with the number of protected attributes.

tributes were selected pseudorandomly (i.e., without tuning) for their problematic associations with protected demographics. Further details on dataset and model hyperparameters are in the Appendix.

**Primary Results.** CelebA-SL and CelebA-ML are depicted in Figures 2, 3 with the key findings:

1. We **can** mitigate intersectional bias and bias amplification, even for hundreds of intersectional groups, using labeled methods like Domain Independent and Weighted ERM.
2. In contrast to prior findings by Liu et al. (2021); Shrestha et al. (2021), we find that unlabeled methods generalize poorly to other fair learning tasks and, even if they perform competitively for small $k$ (number of protected attributes), they have bad intersectional bias for large $k$.
3. There is no consistently superior bias mitigation algorithm. Methods like GDRO, Independent-SP, and Weighted ERM may each perform best depending on the learning task, performance/bias metric, and the value of $k$.

**Intersectional Bias Mitigation** On Celeba-SL (Figure 2), Weighted ERM and GDRO are the best overall methods, with similar accuracy, intersectional bias, and bias amplification for $k \leq 3$. Both methods successfully reduce bias amplification and intersectional bias (versus U-ERM) by a significant margin without losing accuracy. On CelebA-ML (Figure 3), Weighted ERM and Independent-SP are the best overall methods, with Independent-SP outperforming Weighted ERM in terms of bottom quartile mAP (59% vs 53% for $k = 3$) and intersectional bias (5.5 vs 6.2 for $k = 7$). Again, these best methods for CelebA-ML successfully reduce bias amplification and intersectional bias (versus U-ERM) by a significant margin without significant loss of accuracy. These results suggest that bias reduction is possible, even under highly intersectional settings with large $k$, but the optimal choice of a bias mitigation algorithm is context-dependent. All the best methods we found in this section are labeled methods.

**Unlabeled Methods** Prior works found that unlabeled methods perform competitively with labeled methods when protecting a single attribute on CelebA-SL (Shrestha et al., 2021; Liu et al., 2021). Appendix A.1 replicates their experiments with highlights in Tables 2-3. Table 3 confirms that U-JTT yields a competitive worst-case accuracy and bias amplification (85% vs Weighted ERM's 89% accuracy; -0.041 vs -0.055 bias amp). Table 2 shows U-SD also yields competitive worst-case accuracy and bias amplification (82% vs GDRO's 83% accuracy; -0.057 vs -0.065 bias amp).

However, we find these prior results are fragile. Figures 2-3 shows U-JTT has consistently middle-of-the-pack bias scores and low reweighted accuracies for all $k$. Even on Figure 2's $k$=1 case, JTT's 88% reweighted accuracy is 3%+ below others. The only difference between Figure 2's $k$=1 case and Table 3 is that the former protects "Is/Not Pale Skin" attribute instead of "Is/Not Male". In Figure 2 (CelebA-SL), U-SD fares better with competitive reweighted accuracy (92% vs weighted ERM's 92%), bias amplification (0.001 vs weighted ERM's 0.003), and intersectional bias scores (0.3 vs weighted ERM's 0.3) for $k$=1. However, for $k \geq 3$, the intersectional bias of U-SD is

Table 2: Highlights from Shrestha et al. (2021) replication, a variant of Figure 2's CelebA-SL task for $k$=1. A ResNet18 is used and the protected attribute is "Male". U-SD is competitive with GDRO (the best labeled method), and reduces bias significantly versus U-ERM.

| Model | Inter. Bias | Bias Amp | Reweighted Accuracy | Min Accuracy |
|---|---|---|---|---|
| U-ERM | $1.005 \pm 0.109$ | $-0.024 \pm 0.012$ | $0.848 \pm 0.012$ | $0.604 \pm 0.065$ |
| GDRO | $\mathbf{0.558} \pm 0.049$ | $\mathbf{-0.065} \pm 0.008$ | $\mathbf{0.895} \pm 0.009$ | $\mathbf{0.833} \pm 0.034$ |
| U-SD | $0.746 \pm 0.032$ | $-0.057 \pm 0.004$ | $0.887 \pm 0.005$ | $0.817 \pm 0.025$ |

Table 3: Highlights from Liu et al. (2021) replication, a variant of Figure 2's CelebA-SL task for $k$=1. The protected attribute is "Male". U-JTT is competitive with Weighted ERM (the best labeled method), and reduces bias significantly versus U-ERM.

| Model | Inter. Bias | Bias Amp | Reweighted Accuracy | Min Accuracy |
|---|---|---|---|---|
| U-ERM | $1.107 \pm 0.606$ | $0.024 \pm 0.018$ | $0.715 \pm 0.148$ | $0.281 \pm 0.197$ |
| U-JTT | $0.911 \pm 0.005$ | $-0.041 \pm 0.002$ | $0.907 \pm 0.001$ | $0.842 \pm 0.003$ |
| Weighted ERM | $\mathbf{0.647} \pm 0.101$ | $\mathbf{-0.055} \pm 0.012$ | $\mathbf{0.921} \pm 0.003$ | $\mathbf{0.891} \pm 0.026$ |

consistently greater than the labeled baseline, Weighted ERM: 1.9 vs 1.5 for $k = 5$ and 3.0 vs 2.6 for $k = 7$. U-SD was not evaluated on CelebA-ML as its hyperparameters are intractable to tune for large numbers of label classes. These negative results highlight the limitation of unlabeled methods and thus the importance of protected attribute labels, particularly in the multi-attribute domain.

**Intersectional Bias and Bias Amplification Trends** For both CelebA-SL and CelebA-ML, intersectional bias scores are generally increasing in $k$ (see the right-most plots of Figures 2, 3). The noise of the metric itself, however, does not grow significantly with the number of protected attributes. Bias amplification scores are significantly more noisy (see all lines in Figure 2 and the U-JTT line in Figure 3). We also find that bias amplification is uninformative for CelebA-SL with all methods but Independent-SP having a bias amplification score below zero. However, we attribute this to the binary classification nature of CelebA-SL where, although there is class imbalance (85% to 15%), there is a sufficient number of datapoints per class that there is little risk of underfitting with ResNets. Bias amplification remains non-trivial on the multi-label CelebA-ML and, as we will see later, is also non-trivial on the multi-class prediction ImageNet task.

# 5 SCALING TO REAL-WORLD DATASETS

We now consider learning fair models on the ImageNet People Subtree, with protected attribute labels annotated by Yang et al. (2020). This dataset is a realistic benchmark for intersectional bias mitigation, featuring 196 intersectional groups arising from 3 protected attributes: gender, skin color, and age. In addition, only 15,981 images of the dataset's 124,693 images have protected attribute labels—posing a challenge with label scarcity. We found in Section 4 that bias mitigation is possible with hundreds of intersectional groups but that labeled methods are imperative. Can we still effectively mitigate bias on ImageNet with hundreds of protected groups but scarce attribute labels?

We first analyze the performance of existing labeled and unlabeled bias mitigation algorithms on ImageNet. We then improve on those results by proposing a general knowledge distillation based approach (DIR) that helps scale bias mitigation algorithms to the ImageNet People Subtree.

## 5.1 BIAS MITIGATION TECHNIQUES ON THE IMAGENET PEOPLE SUBTREE

**Experimental Setup.** For this section, we use the ImageNet People Subtree annotated by Yang et al. (2020). We have 130,554 images for training (5,861 of which have attribute labels). We have 5,327 images for validation and 5,327 for test, all with attribute labels. 3 protected attributes (gender, skin color, age) are annotated with 196 corresponding intersectional groups. This raises the issue of zero-shot fairness, as 237 out of 284 synsets on the people subtree have zero representation of dark-skinned females (contrast this to the only 3 out of 284 synsets that lack light skinned male representation). The methods considered in this section are as described in Table 1 with the exception of weighted ERM, which we modify to square-root all importance weights–this is to remedy the high-variance importance weights that arise on ImageNet.

**Existing Methods on ImageNet.** Table 4 shows the performance of various labeled and unlabeled methods on ImageNet. The key result is that, using existing labeled bias mitigation methods, we can

Table 4: Results of existing bias mitigation algorithms on the ImageNet People Subtree. Weighted ERM and Independent-SP significantly improve reweighted accuracy and reduce bias amplification versus standard ERM. All intersectional bias scores are within a standard deviation.

|  | Reweighted Acc. | Bias Amp. | Inter. Bias |
|---|---|---|---|
| U-ERM | $45.38 \pm 0.404$ | $0.08153 \pm 0.0095$ | $\underline{2.684} \pm 0.025$ |
| U-JTT | $45.42 \pm 0.460$ | $\underline{0.07603} \pm 0.0105$ | $2.694 \pm 0.006$ |
| U-SD | $36.59 \pm 0.352$ | $0.08489 \pm 0.0116$ | $\mathbf{2.621} \pm 0.027$ |
| GDRO | $45.20 \pm 0.136$ | $0.08910 \pm 0.0170$ | $2.697 \pm 0.014$ |
| Weighted ERM | $\mathbf{47.67} \pm 0.812$ | $0.07611 \pm 0.0011$ | $2.687 \pm 0.027$ |
| Independent-SP | $\underline{47.24} \pm 0.817$ | $\mathbf{0.07402} \pm 0.0095$ | $2.688 \pm 0.049$ |

Table 5: Comparison of pretrained and not-pretrained bias mitigated models on ImageNet. Pretraining significantly improves reweighted accuracy and reduces bias amplification. However, pretraining increases intersectional bias (clarified in Section 5.1).

|  | Reweighted Acc. | Bias Amp. | Inter. Bias |
|---|---|---|---|
| Weighted ERM | $47.67 \pm 0.812$ | $0.07611 \pm 0.0011$ | $2.687 \pm 0.027$ |
| w/o pretrain | $14.88 \pm 0.670$ | $0.09602 \pm 0.0154$ | $2.167 \pm 0.037$ |
| Independent-SP | $47.24 \pm 0.817$ | $0.07402 \pm 0.0095$ | $2.688 \pm 0.049$ |
| w/o pretrain | $17.36 \pm 0.255$ | $0.1429 \pm 0.0110$ | $2.268 \pm 0.012$ |

significantly mitigate bias. Weighted ERM and Independent-SP both outperform standard ERM in terms of reweighted accuracy (47%, 47% vs 45%) and bias amplification (0.076, 0.074 vs 0.082) by a significant margin. The intersectional bias metric is uninformative, i.e., all methods are within a standard deviation of each other.

We also find that unlabeled methods U-JTT and U-SD (45%, 37% accuracy) are significantly outperformed by Weighted ERM and Independent-SP (47% accuracies). U-JTT and U-SD's bias amplification and intersectional bias scores are within a standard deviation of Weighted ERM and Independent-SP's. This corroborates our Section 4 findings of the limited applicability of unlabeled methods in multi-attribute settings where protected attribute information is critical.

**Pretraining.** An important note about the results in Table 4 is that all methods are pretrained with ERM on the entire training set, including data points without attribute labels. This is justified in Table 5 which compares Table 4 results against their not-pretrained counterparts. Pretraining is important for obtaining reasonable performance, improving reweighted accuracy ($15\% \rightarrow 47\%$ for Weighted ERM and $17\% \rightarrow 47\%$ for Independent-SP) and reducing bias amplification ($0.096 \rightarrow 0.076$ and $0.143 \rightarrow 0.074$ respectively). Surprisingly, pretraining increases intersectional bias ($2.16 \rightarrow 2.69$ for Weighted ERM and $2.27 \rightarrow 2.69$ for Independent-SP). This is a flaw of the metric. Low accuracy, high uncertainty neural networks act like a random classifier, which have a intersectional bias close to zero (Eq 4); this is not the case with bias amplification, for instance.

## 5.2 KNOWLEDGE DISTILLATION FOR SCALING LABELED BIAS MITIGATION TECHNIQUES

Our findings in Section 5.1 demonstrate that bias mitigation methods, including Weighted ERM and Independent-SP, can successfully mitigate bias in multi-attribute settings. However, we encountered two key issues from scaling to ImageNet.

**Model Size.** The number of protected groups grows combinatorially with the number of protected attributes. This poses an issue for methods which have a model size linear in the number of protected groups, including Independent-SP, Adversarial, and Discriminative. Their inference complexity increases combinatorially. This necessitated low batch sizes for the experiments in Table 4.

**Overfitting.** Given the large number of protected groups but scarce protected attribute information, labeled bias mitigation algorithms are prone to overfit to available attribute information. For instance, methods which estimate importance weights over protected groups, e.g. Weighted ERM, are prone to high-variance weight estimates. This also affects methods, like Independent-SP and Discriminative, which use protected attribute labels to selectively update network weights: weights associated with underrepresented groups may overfit to the scarce datapoints labeled with the group.

**Knowledge Distillation of Independent Models as a Regularization.** We propose Knowledge Distillation of Independent Models as Regularization (DIR) to address these challenges. Specifically, we develop a two-stage bias mitigation approach, where we distill a set of group-specific

Table 6: Comparison of ImageNet results of using DIR to modify the Weighted ERM and Independent-SP methods. DIR reduces bias amplification while maintaining the reweighted accuracy gains of the original Weighted ERM and Independent-SP methods.

|  | Reweighted Acc. | Bias Amp. | Inter. Bias |
|---|---|---|---|
| Independent-SP | $47.24 \pm 0.817$ | $0.07402 \pm 0.0095$ | $2.688 \pm 0.049$ |
| w/ DIR | $\underline{47.68} \pm 0.521$ | $\mathbf{0.06318} \pm 0.0100$ | $2.693 \pm 0.012$ |
| Weighted ERM | $47.67 \pm 0.812$ | $0.07611 \pm 0.0011$ | $\underline{2.687} \pm 0.027$ |
| w/ DIR | $\mathbf{47.73} \pm 0.920$ | $\underline{0.07354} \pm 0.0031$ | $\mathbf{2.680} \pm 0.015$ |

classifiers into a single classifier. This provides: (1) a bias mitigation method with $O(1)$ inference complexity but the same inductive biases as the $O(k)$ complexity Domain Independent algorithm; and (2) a form of regularization that can be used to augment any labeled bias mitigation algorithm.

Given $G$ protected groups, the Domain Independent (Independent-SP) algorithm (Wang et al., 2020) trains a specialized model $h_g$ for each protected group $g$, training on the same $X \rightarrow Y$ task but exclusively on datapoints labeled with $g$. Given an input $x$, the algorithm directly sums the outputs of the set of $h_g$,

$$z = \sum_{g \in G} h_g(x), \tag{5}$$

where $z$ is final output of the Domain Independent model. Instead of summing over the specialized classifiers $\{h_g \mid g \in G\}$, we propose a distillation-based approach that trains a single classifier $h_d$ with the following loss for each datapoint $(x, y, g) \sim (X, Y, G)$:

$$\ell(h_d(x), y) + \lambda KL(h_d(x), h_g(x)), \tag{6}$$

where $h_g$ are already trained, $\ell(\cdot, \cdot)$ is some loss function and $\lambda$ is the weight for the distillation term. This formulation yields a single classifier $h_d$ that is regularized by the second term in Equation (6), which incentivizes the output $h_d(x)$ to be similar to that of $h_g$, the specialized classifier for the datapoint's protected group $g$. We refer to this scheme as DIR. When $\ell$ is the original task loss function, e.g. cross-entropy, Equation 6 is an approximation of the Domain-Independent algorithm with $O(1)$ rather than $O(k)$ model complexity. However, we can also use DIR to augment other bias mitigation algorithms. For instance, we can augment Weighted ERM by selecting its importance weighted loss as $\ell$. This yields the regularized objective

$$\sum_i w_i \ell(h(x_i), y_i) + \lambda KL(f(x_i), h_{g_x^i}(x_i)), \tag{7}$$

where $w_i$ denotes the importance weight estimate used by Weighted ERM, $g_x^i$ the protected group of $x_i$ and $h_{g_x^i}$ the datapoint's group-specific classifier.

**Experimental Results.** Table 6 depicts the performance gains on ImageNet from using DIR to augment Weighted ERM and Independent-SP. The second and fourth rows correspond to models trained with Equation (6) and Equation (7), respectively. DIR reduces bias amplification for both Independent-SP ($0.074 \rightarrow 0.063$) and Weighted ERM ($0.076 \rightarrow 0.073$) while maintaining the accuracy gains realized by these two labeled methods. Intersectional bias is again uninformative.

## 6 ETHICAL CONCERNS & CONCLUSION

In this paper, we conduct a thorough review of bias mitigation deep learning algorithms in multi-attribute settings. We also propose a knowledge distillation framework for improving bias mitigation on the protected-label-scarce ImageNet dataset. In many applications of machine learning technology that directly or indirectly interface with protected classes, the insights we find concerning multi-attribute fairness may help better inform interventions for bias mitigation.

Due to the nature of the datasets that our experiments use, there are a number of potential ethical concerns arising from the use of protected attribute labels. Dataset schemas used commonly in deep learning literature, such as the subjective "Attractive/Not-Attractive" attribute adopted by the CelebA dataset, can be problematic. In addition, there are privacy and representation concerns inherent in the process of labeling protected attributes. For a more detailed discussion, we refer readers to Yang et al. (2020); Paullada et al. (2020); Birhane & Prabhu (2021) which further explore issues in fair dataset design, especially concerning bias and privacy.

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
