# OpenReview forum: "Scaling Fair Learning to Hundreds of Intersectional Groups"
_ICLR.cc/2022/Conference — ICLR 2022 Submitted_

### Official Review · Reviewer_zw8E · 2021-10-28

**Correctness:** 4
**Technical Novelty And Significance:** 3
**Empirical Novelty And Significance:** 3
**Recommendation:** 6
**Confidence:** 3

**Main Review:**

+ The focus of this paper is an important problem: intersectionality in fairness.
+ The empirical analysis provides insight into how existing methods perform on intersectional groups and highlights fairness challenges that are particularly vexing in the context of intersectionality: runtime complexity and stability.

- The paper would benefit from a discussion on the hyperparameter tuning. Typically this involves a fairness-accuracy tradeoff which would seem to affect the results. e.g, we can always eliminate intersectional bias by returning a constant classifier.
-The current definition of the sensitive attributes invokes the “pale male” default. For clarity and inclusivity, I would recommend revising to define the attribute in terms of the attribute, not the value: e.g, “Gender” instead of “Male.” Are each of these binary? It may also be more appropriate to term these sensitive attributes, since protected typically evokes a legal connotation.
- It would be useful to clarify why we should be concerned about fairness with respect to attributes like “straight hair" or “attractive.” Is this purely illustrative or is there a story the reader should have in mind?
- Accuracy in table 4 seems to be reported on a 0-100 scale but was earlier reported on a 0-1 scale. Why are the reweighed accuracies in Table 4 so low?


Minor:
Define Mean Average Precision.
Define the error bars in the figures. Are all figures for test sets?
Define synsets

**Summary Of The Paper:**


This paper provides an empirical investigation of deep learning bias mitigation methods, focusing on two problems: intersectionality and missing sensitive attribute labels. The paper evaluates several bias mitigation approaches on the Celeb dataset and on ImageNet, the latter having few sensitive attribute labels. The main takeaways from their empirical analysis is that the best performing mitigation method depends on the context and mitigation methods that operate without using the sensitive attribute do not perform well in terms of intersectionality. The paper also proposes a new distillation-based approach to resolve the runtime complexity problem that models like Domain Independent exhibit. On ImageNet their method significantly reduces bias amplification but there are no differences in group-reweighed accuracy or intersectional bias.

**Summary Of The Review:**

This paper provides an extensive empirical analysis of an important question, how fairness methods perform with regards to intersectionality. The paper would benefit from additional discussion on a few points, like hyperparameter tuning.

---

> ### Author Response · Authors · 2021-11-22
> **Thank you for your comments!**
>
> Thank you for reviewing and your detailed feedback! We offer some clarifications for your specific comments below.
>
> *A discussion on the hyperparameter tuning, maybe involving a fairness-accuracy tradeoff.*
>
> Some methods, such as group DRO, do not give an obvious way to tune fairness-accuracy trade-offs. Other methods, such as invariant risk minimization, have parameters that may weakly relate to the fairness-accuracy tradeoffs. We tuned the hyperparameters in our results to maximize a common accuracy metric for a more apples-to-apples comparison between methods, but we believe that a broader discussion of hyperparameter tuning for these various methods would be limited by the limited tunability of certain methods and out of the scope of our intersectional fairness contributions. However, we will revise our discussion of hyperparameters in the Appendix to characterize how each hyperparameter choice relates to fairness-accuracy tradeoffs.
>
> *The current definition of the CelebA sensitive attributes invokes the “pale male” default.*
>
> The current definition of the sensitive attributes, including “Pale” and “Male, are the definitions used by the original CelebA dataset creators. We have retained the original attribute terms, e.g., Is-Male rather than Gender, for continuity with prior works, but completely agree that this may be undesirable. We will update our manuscript to discuss the concerns raised by the current labeling scheme, including the non-binary nature of many of these attributes.
>
> *Is our concern about fairness with respect to attributes like “straight hair" or “attractive” purely illustrative?*
>
> This is purely illustrative.
>
> *Why are the reweighed accuracies in Table 4 so low?*
>
> The numbers in Table 4, including the reweighted accuracies, look low because we adopt a difficult ImageNet task. We chose a difficult subset of the ImageNet dataset, the People subtree.  This is not the usual subset adopted by other computer vision literature, which instead generally use the ILSVRC subset. The closest existing numbers for our dataset split are those of the Yang 2019 paper we cite, who also use the People tree of the ImageNet dataset. They obtain a top-1 (unweighted) accuracy of ~56%, predicting from 143 classes. We obtain a top-1 (unweighted) accuracy of ~48-50%, predicting from 284 classes. In short, our baseline accuracies are in line with what one would expect for the ImageNet version that we use.
>
> *Minor: Define Mean Average Precision. Define the error bars in the figures. Are all figures for test sets? Define synsets*
>
> Thank you for these suggestions; we will correct these omissions in our revision. For clarification, all figures are for test sets and error bars always denote standard deviation.

---

### Official Review · Reviewer_JEnE · 2021-10-29

**Correctness:** 3
**Technical Novelty And Significance:** 3
**Empirical Novelty And Significance:** 4
**Recommendation:** 6
**Confidence:** 3

**Main Review:**

Intersectional fairness is a practical and important problem. The authors conduct a comprehensive comparison among existing bias mitigation techniques, which I find valuable. The paper begins reasonably well-written but the clarity deteriorates a bit in Section 3.

1. My main concern is the fairness metrics considered in this paper.

First, the authors introduced a notion called worst-case accuracy in Eq. (2) and argued that it was a surrogate metric for equalized odds. However, it is unclear to me why it is an appropriate surrogate. Note that equalized odds measure the *performance disparity* instead of the worst-case accuracy.

Second, it is unclear to me why the authors computed the surrogate metrics instead of the original measures (e.g., demographic parity) when analyzing the performance of bias mitigating algorithms (see e.g., Fig 2). I understand that training a model with the original fairness measures can be hard. However, once a binary classifier is fixed, verifying demographic parity is just to computing the mean of Bernoulli variables.

Third, the definitions of the fairness metrics in Section 3.1 are puzzling from a mathematical perspective.

(1)  the importance-weighted accuracy in Eq. (1) involves 1/|g| inside the expectation. It is unclear what |g| means. Is it the maximal number of subgroups? If so, then Acc_U(h) -> 0 when |g| -> \infty, which does not make any sense.

(2) the bias amplification in Eq (3) involves Pr(g|h(x)). This conditional probability distribution is unclear to me. First, is h(x) a random variable? If so, the authors should write it as h(X) to be consistent. Second, if h(x) is a random variable, why does the measure s_y only rely on y?

(3) the intersectional bias in Eq (4) is also unclear. Isn’t it \max \min instead of \max \argmin?

I highly suggest the authors carefully revise their definitions of the fairness metrics in Section 3.1!

2. The primary results in Section 4 are interesting. I wonder if the authors can provide any interpretation of these results.

**Summary Of The Paper:**

This paper conducts a comprehensive comparison among existing bias reduction methods when there are multiple protected attributes. The authors also propose a knowledge distillation framework for improving fairness on the protected-label-scarce ImageNet dataset.

**Summary Of The Review:**

Interesting experimental results but the fairness metrics adopted in this paper need to be clarified

---

> ### Author Response · Authors · 2021-11-22
> **Thank you for your comments!**
>
> Thank you for reviewing and your detailed feedback! We offer some clarifications for your specific comments below.
>
> *Worst-case accuracy as a surrogate metric for equalized odds.*
>
> We agree that worst-case accuracy is not an exact proxy for equalized odds. We intended to communicate that the two measures are similar in spirit, as they both examine the model’s performance on, or between, specific protected groups. This is in contrast to the demographic parity and bias amplification measures, which do not examine the model’s performance/accuracy and rather the model’s predictions. We will revise our manuscript to clarify that we intended to contrast metrics which relate to performance versus metrics which relate to outcomes/predictions.
>
> *Use of surrogate metrics instead of original measures (e.g., demographic parity).*
>
> Our choices of surrogate metrics were generally motivated by the following main reasons.
> 1. The surrogate metrics we use in our paper are standard for empirical fairness papers. To our knowledge, most of the related papers we cite on deep learning fairness use only a strict subset of the performance metrics we present. Our selection of performance metrics is more broad than any of our closely related works, which also replace classical  measures with more reliable surrogates (Wang et al., Shrestha et al., 2021, Liu et al., 2021).
> 2. Our surrogate metrics were usually equivalent under transformation, or similar in spirit, to their original measure. For example, demographic parity is equivalent to our intersectional fairness metric (Foulds et al., 2020).
> 3. However, even though our metrics may be equivalent under transformation, said transformations affect how the metrics are averaged across different protected groups. In intersectional settings, there are many protected groups and so the averaging behavior of metrics is critical (e.g., by affecting interpretability/noise). We originally deferred justifications of these surrogates to the prior works like Zhao et al., 2017, but will revise to include additional commentary.
>
> *The importance-weighted accuracy definition in Eq. (1).*
>
> The term |g| denotes the number of datapoints corresponding to a protected group. By weighting datapoints by the inverse of |g|, we weight a datapoint by how represented its protected group is in the dataset.
>
> *The bias amplification definition in Eq (3).*
>
> Thank you for catching this. h(x) and y are both random variables. h(x) is the label predicted for datapoint x, and should have been written as the expectation E[h(x)], where the expectation is taken over the datapoints x. We will amend our definition.
>
> *The intersectional bias definition in Eq (4).*
>
> Thank you for catching this. Yes, it should have been \min_\epsilon \epsilon instead of \argmin_\epsilon. We will amend this equation as well.

---

### Official Review · Reviewer_CCHj · 2021-11-02

**Correctness:** 3
**Technical Novelty And Significance:** 2
**Empirical Novelty And Significance:** 2
**Recommendation:** 5
**Confidence:** 3

**Main Review:**

Strength

1. The empirical analysis of existing bias mitigation methods on two large datasets with multiple sensitive attributes/partially labeled protected attributes is very interesting. It shows the scalability of existing methods and the potential challenges when applying those methods to large-scale datasets.

2. A new method for group-specific models is proposed to augment other mitigation algorithms.


Weakness:

1. The novelty of this work concerns me the most. The empirical analysis is interesting but far from the expectation of the standard of ICLR. In addition, the proposed DIR method is incremental for model simplification of bias mitigation. It would be considered as transferring the existing method **Domain-Independent model** into a smaller one whose inference complexity is less than the original version, through incorporating the knowledge distillation approach. The distillation process requires |G| pre-trained models for |G| groups.

2. The claimed O(1) complexity is implausible because it assumes the availability of G group-specific models. It seems the proposed method is not able to reduce the complexity without G models. ~~In addition, it is unclear how to integrate multiple classifiers $h_g$ into Eq (6) and (7) because $\lambda$ is not model-specific.~~ (Resolved)

3. It is unclear how to integrate the DIR method with unlabeled mitigation algorithms as the empirical results show they are vulnerable to poor generalization.


**Summary Of The Paper:**

This paper conducts an empirical analysis of existing bias mitigation methods on two large datasets CelebA and ImageNet where there are multiple sensitive attributes and some unavailable protected labels. The results show the existing can mitigate intersectional bias at scale but the unlabeled methods generalize poorly. This paper further proposes a knowledge distillation of independent models as regularization method (DIR) which is able to augment into other bias mitigation algorithms.

**Summary Of The Review:**

The empirical analysis is interesting but the incremental contribution concerns me the most. In addition, the proposed DIR method is trivial and contains many uncertainties. I would not root for this paper.

---

> ### Author Response · Authors · 2021-11-22
> **Thank you for your comments!**
>
> Thank you for reviewing and your detailed feedback! We offer some clarifications for your specific comments below.
>
> *The O(1) complexity of group-specific models.*
>
> We appreciate the chance to clarify this comment. We claim that our knowledge distillation approach has an inference runtime that is independent of the number of protected attributes. This is in contrast to existing methods whose inference runtime can scale combinatorially with the number of protected attributes. Our knowledge distillation approach does require training and storing G group-specific models. However, in production environments, inference complexity is generally the bottleneck rather than storage or training complexity.
>
> *The integration of multiple classifiers into Eq (6) and (7).*
>
> We use lambda to weigh the importance of the distillation loss in the second term compared to the loss in the first term. We assume the access to the group label g, for a data point x, and thus can select the corresponding h_g for x in Eq (6) and (7).
>
> *The integration of the DIR method with unlabeled mitigation algorithms.*
>
> Our formulation of knowledge distillation requires access to group labels, and so cannot be integrated with unlabeled mitigation algorithms. The poor generalization behavior of unlabeled mitigation algorithms is, however, a motivation for our knowledge distillation approach. Unlike unlabeled mitigation algorithms, the computational complexity and learning dynamics of labeled mitigation algorithms depends on how intersectional groups are labeled. This motivates our knowledge distillation approach which allows labeled algorithms to scale well despite having many intersectional groups to deal with.

---

### Official Review · Reviewer_F8SZ · 2021-11-03

**Correctness:** 2
**Technical Novelty And Significance:** 2
**Empirical Novelty And Significance:** 2
**Recommendation:** 5
**Confidence:** 4

**Main Review:**

### Strengths

- Focus on an important and understudied problem.

- Paper presents a new approach to handle intersectional attributes: Knowledge Distillation of Independent Models as Regularization (DIR).

### Weaknesses

- [Major] Paper evaluates methods on datasets where we have no ground truth. These findings should be confirmed on a synthetic dataset where we have ground truth labels (e.g., binary classification task with $k = 1, 2, 4$ binary group attributes).

- [Major] Key findings of the empirical study are based on summary statistics that do not capture or reflect differences in accuracy across intersectional groups (see e.g., Figure 1, Figure 3, Tables 2 and 3). For example:

     (*) The ``reweighted accuracy" with k = 128 groups does account for the performance of all intersectional groups. However, it would be unable to flag instances where an intersectional group has an accuracy of 0\% since any group affects at most 1/128 of the total performance.

     (*) Measures such as Bias Amplication and "Intersectional Bias Score," are surrogate measures that suffer from their own issues (as discussed by the authors on Page 4). The informative-ness of these issues is also hindered in that they represent population-level statistics.

      I would recommend would be better to evaluate methods in ways that do not involve summary statistics at all (e.g., via a plot showing the distribution of group-specific accuracy). I would also recommend the authors use measures of group-level performance (e.g., worst-case accuracy over groups, # of groups with substantial bias).

- [Minor]: The paper is missing references to some of the earlier work on this topic, namely: (1) work on training single classifiers with fairness guarantees over an infinite number of intersectional groups [Kearns et al - ICML 2018](https://proceedings.mlr.press/v80/kearns18a.html); (2) work on decoupled training across many intersetional attributes (see e.g., [Ustun et al at ICML 2019](http://proceedings.mlr.press/v97/ustun19a.html)).

**Summary Of The Paper:**

This paper studies the fairness of deep learning models in classification tasks with a large number of intersectional groups. The paper has two primary contributions:

1. The paper includes an empirical study that shows the inherent difficulty of this setting (i.e., due to the lack of labels), and the limitations of existing approaches.

2. The paper presents a new approach to train deep models to settings with many intersectional groups called Knowledge Distillation of Independent Models as Regularization (DIR).

**Summary Of The Review:**

This paper studies an important problem. In short, there is little work on intersectional subgroups, even less on deep learning. In this context, I see the empirical study as an opportunity to make a contribution as it can highlight previously unknown issues.

My recommendation is based on the following issues: (1) key findings are not well-supported by the empirical results (see comment above); (2) the proposed approach DIR is underdeveloped and under-evaluated in the text. I think that both issues could be addressed, but should be subject to another round of peer review.

---

> ### Author Response · Authors · 2021-11-22
> **Thank you for your comments!**
>
> Thank you for reviewing and your detailed feedback! We offer some clarifications for your specific comments below.
>
> *Flagging instances where an intersectional group has an accuracy of 0%.*
>
> We do flag groups with 0% accuracy with our “worst-case group accuracy” metric. However, we generally defer this metric in favor of “reweighted accuracy” because worst-case accuracy is too pessimistic in intersectional settings. With a combinatorial number of protected groups, there are often protected groups with only a single datapoint represented in the test set. This means that a 0% worst-case group accuracy is near-inevitable in intersectional settings, and not an informative metric. Summary metrics like re-weighted accuracy allow us to at least obtain some signal for performance, despite the difficulties of evaluating in these settings.
>
> *Confirmation on a synthetic dataset with ground truth labels.*
>
> As previous works have already evaluated fairness in toy synthetic environments including Colored MNIST and Biased MNIST, we focused on real-world large-scale image recognition settings on ImageNet and CelebA where we took the dataset labels as ground-truth.
>
> *Missing references.*
>
> Thank you for these references. We are adding additional commentary on these works in our revision.

---

### Author Response · Authors · 2021-11-22
**General Comment**

We thank all reviewers for their thoughtful feedback. We will post a revision to incorporate your comments.
A high-level summary of the feedback:
* All reviewers agree that intersectional fairness is a practical and important problem.
* The experiments are comprehensive and yield interesting insights (zw8E, JEnE, CCHj).
* The fairness metrics adopted in this paper need to be clarified. In particular, the reason for choosing summary (F8SZ) and surrogate (JEnE) statistics.

To address F8SZ and zw8E, we discuss below why we believe our key contributions offer significant insights for the rest of the community. We have also provided individual responses to your specific comments.
1. Our empirical study shows how the unlabeled bias mitigation techniques touted in recent literature have serious deficiencies in multi-attribute settings. We also demonstrated the inconsistent performance of state-of-art labeled bias mitigation algorithms, and found that naive importance weighting was one the most reliably effective methods. We believe these insights contribute to the fair ML community's discussion around fair deep learning by highlighting and addressing important issues with existing algorithms.
2. We agree that the knowledge distillation technique we introduce, DIR, is straightforward and its design. However, we felt that the effectiveness of this simple technique, both in terms of performance and inference complexity,  warranted DIR's inclusion in our paper. DIR is an intuitive first-step towards addressing the concerns we raise in the core of our paper.

In addition, we have also posted a set of additional experiments to the revised Appendix. These experiments repeat our core empirical analysis but for different choices of protected attributes. The findings clarify that the trends we identify in the main body of our paper hold generally, even when protected attributes are particularly noisily labeled, balanced in label distribution, or imbalance in label distribution.

---

### Decision · Program_Chairs · 2022-01-20

**Decision:**

Reject

**Comment:**

The manuscript performs an empirical analysis of existing bias mitigation methods on two large datasets CelebA and ImageNet People Subtree where there are multiple sensitive attributes and some unavailable sensitive attribute labels. The results show that existing methods can mitigate intersectional bias at scale but unlabeled mitigation methods generalize poorly. The manuscript further proposes a knowledge distillation approach which can augment other labeled mitigation approaches.

On the positive aspect, the manuscript studies an important problem: intersectional subgroups on deep learning methods. Reviewers acknowledged that an empirical study on this problem is as an opportunity to make a contribution as it can highlight previously unknown issues.

There are however several major concerns including:
1. Methodological contribution (knowledge distillation) is under-developed, while empirical investigation is interesting but can be further developed;
2. The fairness metrics adopted in this manuscript need to be clarified;
3. A discussion on the hyperparameter tuning, maybe involving a fairness-accuracy tradeoff;
4. The claimed O(1) complexity for the knowledge distillation approach is implausible because it assumes the availability of G group-specific models. This has been clarified in the rebuttal that the claim is only for the inference complexity, and the approach does not improve the training complexity.

Reviewers also concluded that while the empirical analysis is interesting, the results on CelebA to be of limited use because the sensitive attributes are "purely illustrative." It's not clear that the insights from these illustrative intersectional groups (e.g. big nose & attractive) will hold for groups that are meaningful in a fairness sense.